# Low-Dose-Rate Radiation-Induced Secretion of TGF-β3 Together with an Activator in Small Extracellular Vesicles Modifies Low-Dose Hyper-Radiosensitivity through ALK1 Binding

**DOI:** 10.3390/ijms23158147

**Published:** 2022-07-24

**Authors:** Ingunn Hanson, Kathinka E. Pitman, Ursula Altanerova, Čestmír Altaner, Eirik Malinen, Nina F. J. Edin

**Affiliations:** 1Department of Physics, University of Oslo, 0371 Oslo, Norway; k.e.pitman@fys.uio.no (K.E.P.); eirik.malinen@fys.uio.no (E.M.); n.f.j.edin@fys.uio.no (N.F.J.E.); 2Department of Stem Cell Preparation, St. Elisabeth Cancer Institute, 84505 Bratislava, Slovakia; ursula.altanerova@ousa.sk (U.A.); cestmir.altaner@ousa.sk (Č.A.); 3Cancer Research Institute, Slovak Academy of Sciences, Bratislava, 94505 Bratislava, Slovakia; 4Department of Medical Physics, Oslo University Hospital, 0379 Oslo, Norway

**Keywords:** ADAM, adaptive response, ALK1, ALK5, bystander effects, hyper-radiosensitive response, low-dose radiation, low dose rate, MMP, TGF-β3

## Abstract

Hyper-radiosensitivity (HRS) is the increased sensitivity to low doses of ionizing radiation observed in most cell lines. We previously demonstrated that HRS is permanently abolished in cells irradiated at a low dose rate (LDR), in a mechanism dependent on transforming growth factor β3 (TGF-β3). In this study, we aimed to elucidate the activation and receptor binding of TGF-β3 in this mechanism. T-47D cells were pretreated with inhibitors of potential receptors and activators of TGF-β3, along with addition of small extracellular vesicles (sEVs) from LDR primed cells, before their radiosensitivity was assessed by the clonogenic assay. The protein content of sEVs from LDR primed cells was analyzed with mass spectrometry. Our results show that sEVs contain TGF-β3 regardless of priming status, but only sEVs from LDR primed cells remove HRS in reporter cells. Inhibition of the matrix metalloproteinase (MMP) family prevents removal of HRS, suggesting an MMP-dependent activation of TGF-β3 in the LDR primed cells. We demonstrate a functional interaction between TGF-β3 and activin receptor like kinase 1 (ALK1) by showing that TGF-β3 removes HRS through ALK1 binding, independent of ALK5 and TGF-βRII. These results are an important contribution to a more comprehensive understanding of the mechanism behind TGF-β3 mediated removal of HRS.

## 1. Introduction

The low dose region of ionizing radiation (IR) is of concern in cancer radiotherapy, especially through inevitable irradiation of healthy tissue in the radiation field. The extensive use of fractionated radiotherapy calls for an understanding of the mechanisms induced by repeated low-dose exposures. An increased awareness of how these mechanisms differ between malignant and healthy tissues may lead to the discovery of approaches to manipulate the response of these tissues in order to increase the sensitivity of the former, or the resistance of the latter.

When irradiated with doses below approximately 0.5 Gy, about 80% of tested cell lines exhibit a hyper-radiosensitive (HRS) response to X-ray, γ-ray, proton, carbon ion, and pi-meson irradiation [1,2,3,4,5,6]. It is well documented that subjecting HRS-competent cells to an acute, high-dose-rate (HDR) priming dose of 0.2–0.3 Gy transiently removes the HRS response to subsequent challenge irradiation [7,8,9,10,11]. In these cases, HRS is recovered within about 48 h. We have previously shown that giving the same priming dose at a low dose rate (LDR) of approximately 0.3 Gy/h permanently removes the HRS response from the cells and their progeny [7]. Transforming growth factor β (TGF-β) is a family of pleiotropic proteins that are ubiquitously expressed in mammalian tissue. The three family members that are found in humans, TGF-β1–3, have high structural similarity and exercise both overlapping and separate functions in a range of physiological and pathological processes including wound healing and fibrosis, inflammation and cancer [12].

Proteins of the TGF-β family signal through a heteromeric complex with two transmembrane serine/threonine receptors. TGF-β first binds to a type II receptor, which then recruits and activates a type I receptor [13]. There are seven known type I receptors (Activin receptor-like kinase (ALK) 1–7) and five type II receptors (ActRIIA, ActRIIB, BMPRII, TGF-βRII and AMHRII) [14]. Generally, ALK4, ALK5 and ALK7 bind to TGF-β, whereas ligands for ALK1, ALK2, ALK3 and ALK6 belong to the bone morphogenic protein (BMP) and growth differentiation families of proteins [15]. The TGF-βRII/ALK5 receptor complex is the best described signaling system for all three TGF-βs. The bound and activated receptor complex phosphorylates Smad proteins to propagate the signal [13].

TGF-β family members are transcribed as proproteins and remain inactive after initial intracellular cleaving by furin-type enzymes because of an unusually high affinity between the propeptide latency-associated peptide (LAP) and TGF-β [12]. Furthermore, TGF-β-LAP is often found as a tricomplex with the latent TGF-β binding protein (LTBP), termed the large latent complex (LLT). The LLT is capable of binding to extracellular matrix (ECM) components and thereby act as a depository of TGF-β.

Activation of TGF-β requires separation from LAP and is necessary for binding to the TGF-β receptor complex. Separation and subsequent activation may follow from diverse processes such as exposure to acidic or basic solutions, heat and reactive oxygen species (ROS). A number of biological molecules, including several members of the metzincin superfamily, also have the ability to activate TGF-β family members [16,17,18,19]. The metzincin superfamily of endopeptidases consists of 23 matrix metalloproteinases (MMPs), 13 disintegrin metalloproteinases (ADAMs) and 19 ADAMs with thrombospondin motifs (ADAMTS). In addition to activation of TGF-β, members of the metzincin superfamily have a large variety of targets that in turn mediate a range of cellular mechanisms such as cellular differentiation and proliferation, cell motility and structural stability of the ECM [20].

In previous publications, we showed that cell medium that had been conditioned by LDR primed cells and later transferred to unirradiated reporter cells transiently removed HRS in the reporter cells upon subsequent challenge irradiation [21]. We also showed that medium conditioned by unirradiated cells that was collected and LDR irradiated without cells present affected the reporter cells in an equivalent manner [22]. Both mechanisms were TGF-β3 dependent, as neutralizing TGF-β3 antibody retained HRS in the reporter cells. Together, these results suggested that TGF-β3 is secreted from LDR-primed and unirradiated cells and needs to be activated by LDR irradiation in order to remove HRS. The exact nature of the activation process is still unclear, due to the above-mentioned variety of potential activation methods for TGF-β.

In this study, we aimed to elucidate the process of TGF-β3 activation and transfer from LDR primed cells to unirradiated reporter cells. We show that the factors in the cell medium responsible for removing HRS are released from LDR irradiated cells in small extracellular vesicles (sEVs). We also show that sEVs contain TGF-β3 regardless of irradiation, indicating an irradiation-dependent activation of TGF-β3 in sEVs from LDR-primed cells. We demonstrate that TGF-β3 is activated by one or several members of the MMP or ADAM protein families in LDR irradiation-induced removal of HRS. In the reporter cells, we show that the TGF-βRII/ALK5 receptor complex is not responsible for HRS removal. We identify for the first time a functional interaction between TGF-β3 and ALK1 when we show that removal of HRS is dependent on TGF-β3 binding to ALK1. We also uncover a competition between ALK5 and ALK1 for the ligand, where ALK5 demonstrates the higher affinity, whereas ALK1 mediates the mechanism of HRS abolition.

## 2. Results

### 2.1. sEVs from LDR Primed Cells Removed HRS in Reporter Cells

To investigate whether the radioprotective effect of TGF-β3 was transferred through sEVs, these were isolated from LDR primed T-47D cells or unirradiated T-47D control cells and added to the medium of unirradiated reporter T-47D cells. The radiosensitivity of the reporter cells was then analyzed via the clonogenic survival assay. Isolation of sEVs was verified by transmission electron microscopy (TEM) imaging and the presence of proteins specific to sEVs detected by mass spectroscopy [23] (Appendix A). While sEVs from control cells did not influence the radiosensitivity of reporter cells, sEVs from LDR primed cells altered their survival curve to that consistent with removal of HRS (Figure 1a). This effect is similar to the direct LDR irradiation and transfer of medium from LDR primed cells (irradiated cell-conditioned medium, ICCM) (Figure 1b). 

We confirmed that removal of HRS by sEV transfer was dependent on TGF-β3 by adding a neutralizing antibody of TGF-β3 to the sEVs during transfer, a process that retained the HRS response in the reporter cells (Figure 1c). The involvement of inducible nitric oxide synthase (iNOS) was confirmed by addition of iNOS inhibitor 1400 W, which negated the effect of sEVs from LDR primed cells.

### 2.2. sEVs Contained TGF-β3 Regardless of LDR Priming

To elucidate the mechanism of HRS removal by sEVs from LDR primed cells, we analyzed their proteomic profile and compared it to that of sEVs from unirradiated control cells. In light of the results from the proteomic analysis, we re-analyzed mRNA quantification of LDR primed and unirradiated T-47D cells from a previous published study [24]. TGF-β3 was detected in both groups of sEVs, with no significant difference between groups (Figure 2). Neither TGF-β1 nor TGF-β2 was detected in any of the sEV samples. In the mRNA analysis, TGF-β3 was slightly upregulated in the LDR primed cells, while there was no detectable difference in TGF-β1 or TGF-β2 levels. 

Due to the nature of the detection technique (tandem mass spectrometry (MS/MS) with trypsin digestion of the samples), it was not possible to determine whether the detected TGF-β was attached to its propeptide LAP, or if it was in an active state, as fragments from both sections of the protein was detected in sEV samples (Appendix A). This, together with the fact that sEVs from LDR primed cells removed HRS in reporter cells, led us to explore the possibility that TGF-β3 was secreted in a latent form bound to LAP in sEVs from both LDR primed and control cells, possibly together with an activating protein in the LDR primed cells.

A total of 14 proteins were differentially detected with a *p*-value of <0.05 (one-way ANOVA) between the sEVs from LDR primed cells and those from controls (Figure 3). To look for potential activators of TGF-β3 among these, we performed a protein-protein interaction (PPI) analysis, using the STRING database [28], of differentially detected proteins with potentially relevant query proteins, such as the TGF-β family and their latent binding proteins, potential receptors for TGF-β3, iNOS and its relatives, and interleukin 13 (IL13) (Figure 4, Appendix A). Metalloproteinase inhibitor 1 (TIMP1) was identified by STRING as a potential interactor of several of the query proteins, including TGF-β3. 

TIMP1 is one of four mammalian metalloproteinase inhibitors with similar inhibitory functions, but varying affinity for specific ligands [20]. While TIMP1 was differentially detected among the proteins in sEVs from LDR primed and control cells, TIMP2 and TIMP3 were significantly down- and upregulated, respectively, in terms of mRNA in LDR primed whole cells, compared to unirradiated controls (Figure 2). Several of the ligands of TIMPs, metalloproteinases of the metzincin superfamily, have been shown to activate TGF-β family members [16,17,18,19]. In the sEVs from LDR primed and control cells, ADAM9 was detected in similar amounts in both groups. In mRNA from LDR primed and control whole cells, 6 MMPs, 6 ADAMs (including ADAM9) and 6 ADAMTSs were detected, most of which were significantly up- or downregulated between the groups (Figure 2). 

### 2.3. Inhibition of Metalloproteinases Retained HRS in Reporter Cells 

To assess the involvement of MMP, ADAM or ADAMTS proteins in the activation of TGF-β3 in ICCM- and sEV-mediated removal of HRS in reporter cells, we used a broad-spectrum MMP/ADAM inhibitor, TAPI-2, which was added to the medium of LDR primed T-47D cells. After conditioning of the medium, the cells were filtered out, and the ICCM with TAPI-2 was transferred to unirradiated T-47D reporter cells. The reporter cells were then cultivated for 24 hours before re-seeding in fresh medium and subjected to challenge irradiation. Indeed, when metalloproteinases were inhibited in the ICCM, the resulting surviving fraction after 0.1 Gy x-irradiation was consistent with HRS, and lower than that after transfer of ICCM alone, which was similar to that after LDR priming of the cells (Figure 5a). Pretreatment with TAPI-2 alone, without transfer of ICCM, did not influence the surviving fraction of cells that had been irradiated with 0.1 or 0.3 Gy (Figure 5b). It is, therefore, likely that one or several members of the MMP or ADAM protein families are responsible for activation of TGF-β3 in this mechanism.

### 2.4. TGF-β3 Binds to ALK1 to Remove HRS in Reporter Cells

To elucidate TGF-β3 mediated removal of HRS in cells, we wanted to identify the receptor involved in this mechanism. While the TGF-βRII/ALK5 receptor complex is the best described for all three TGF-β isoforms, ALK1 has been identified as an alternative receptor for TGF-β1/TGF-βRII, and TGF-β3 has been assigned as a ligand to ALK1 [29,30]. We first tested whether ALK5 was involved in the function of TGF-β3 on removal of HRS by adding an ALK5 inhibitor together with TGF-β3 to cells prior to challenge irradiation. The resulting survival curve was consistent with removal of HRS (Figure 6a and Figure 7a), indicating that TGF-β3 mediates its radioprotection by binding to another receptor. Next, we used K02288, a selective type BMP receptor inhibitor with effect on ALK1, ALK2 and ALK6, to investigate the involvement of these three receptors, and found that HRS was indeed removed (Figure 6b and Figure 7b). To exclude the involvement of ALK2 and ALK6, we tested LDN193189, a blocker of ALK2, ALK3 and ALK6 at a concentration of 10 µM, and found that this did not affect the ability of TGF-β3 to remove HRS in reporter cells. Interestingly, inhibition of TGF-βRII with a neutralizing antibody had no effect on removal of HRS by TGF-β3. Thus, TGF-β3 appears to remove HRS through ALK1 by a mechanism independent of TGF-βRII.

### 2.5. Inhibition of ALK5 Leads to Removal of HRS without Addition of TGF-β3

Surprisingly, inhibition of ALK5 without addition of TGF-β3 modified the radiation response in reporter cells in a manner that was consistent with the addition of TGF-β3 (Figure 7a,c). When neutralizing TGF-β3 antibody was added together with the ALK5 inhibitor, HRS was not removed, proving that TGF-β3 was essential in this mechanism. iNOS inhibitor 1400 W also negated the effect of TGF-β3 on the survival fraction after 0.2 Gy (Figure 7d). Similarly, inhibition of ALK1 and ALK5 together did not remove HRS. Together, these findings point to a competition between ALK1 and ALK5 for the ligand TGF-β3, where ALK5 has the higher affinity, but ALK1 modifies the radiation response. 

Peptidyl-prolyl cis-trans isomerase FKBP4 (FKBP4), which was upregulated in sEVs from LDR primed cells, is known to associate with heat shock protein 90 (HSP90) to form steroid receptor heterocomplexes [31]. HSP90 has in turn been shown to interact with and stabilize ALK5 [32,33,34], and therefore, we considered the possibility that an increase in the FKBP4 concentration in LDR primed cells led to a decrease in the amount of HSP90 available to stabilize ALK5, leading to a condition similar to inhibition of ALK5 and thereby removal of HRS. However, addition of recombinant FKBP4 did not alter the radiosensitivity of T-47D cells or remove the HRS response (Appendix A).

## 3. Discussion

### 3.1. sEVs Transport the Radioprotective Factor(s)

We showed here that sEVs from LDR primed T-47D cells removed the HRS response to low-dose challenge irradiation in previously unirradiated reporter cells. This effect was equivalent to the one that was previously observed after pretreatment with LDR ICCM [21]. We also showed that removal of HRS via pretreatment with sEVs was dependent on TGF-β3 and iNOS, as neutralizing antibody to TGF-β3 or iNOS inhibitor 1400 W retained HRS in the reporter cells. The roles of TGF-β3 and iNOS in removal of HRS via pretreatment with ICCM were previously established [24,35]. On this basis, we propose that removal of HRS with transfer of ICCM or sEVs is the same process, governed by sEVs in the cell medium.

It is possible that the active factor(s) that removes HRS is secreted both in sEVs and via other secretion pathways. However, sEVs and other extracellular vesicles (EVs) provide a method of communication between cells where a specific selection of proteins, lipids and nucleic acids can be packaged and delivered together to the recipient cell. This could be the transfer mechanism for the factors necessary for removal of HRS. We have previously demonstrated that these factors include TGF-β3, iNOS, IL13 and peroxynitrite (ONOO^−^) [24,35,36], and in the present study, we have shown that a member of the MMP or ADAM protein families is included as activator of TGF-β3. Several studies have shown sEVs and other EVs to be mediators of radiation-induced bystander effects in vivo [37] and in vitro [38,39]. AL-Mayah et al. observed that sEVs from unirradiated cells that exhibited bystander effects via sEVs transfer could in turn induce bystander effects in other unirradiated cells [40]. 

In previous studies, we concluded that TGF-β3 was the active factor in ICCM that removed HRS in reporter cells. However, the HRS response was also removed from recipient cells by taking medium from unirradiated cells and LDR irradiating the medium without cells present with 0.3 Gy at 0.3 Gy/h [24], indicating a possibility of post-secretion activation of TGF-β3. Thus, either the unirradiated control cells secrete TGF-β3 in an inactive complex with the propeptide LAP, while the LDR primed cells secrete the activated form dissociated from LAP, or more likely, the LDR primed cells secrete the inactive complex with LAP together with something that induces activation.

TGF-β3 was detected in sEVs from both LDR primed and unirradiated cells. During MS/MS analysis, peptide fragments from a partially digested protein were counted and used to identify the protein. Digestion by trypsin cut the peptide into fragments next to the amino acids lysine (K) or arginine (R). When analyzing the content of sEVs from LDR primed and control T-47D cells, peptide fragments from both the propeptide LAP section (amino acid 1–300) and the mature TGF-β3 section (amino acid 300–412) of the TGF-β3 molecule were detected in similar amounts in both groups. The last amino acid in the LAP section was arginine, which is a potential cleavage site, even if TGF-β3 was initially bound to LAP. It was, therefore, impossible to determine from the MS/MS analysis if the detected TGF-β3 was in its active or latent form. However, since sEVs from unirradiated control cells did not remove the HRS response in reporter cells, and sEVs from LDR primed cells only did so when TGF-β3 was present and active, it is reasonable to believe that TGF-β3 is the active factor in removal of HRS by transfer of sEVs from LDR primed cells. This supports the claim that TGF-β3 is produced and secreted in sEVs regardless of priming irradiation, and that it is activated by LDR irradiation. In this case, several potential activation mechanisms should be considered, one of them being activation via endopeptidases or other proteins.

### 3.2. Metalloproteinase Activation of TGF-β3

TIMP1, which was found to be significantly upregulated in sEVs from LDR primed cells, was suggested to interact with TGF-β1–3, ALK5, IL13, and endothelial nitric oxide synthase, which shares its main function of nitric oxide production with iNOS. TIMPs are inhibitors of selected MMP/ADAM/ADAMTS proteins, which are possible activators of TGF-β [12]. While TIMP1 was upregulated in LDR sEVs, TIMP2 mRNA was found to be significantly downregulated in LDR primed whole cells. This led us to explore the possibility that a shift in the balance of TIMP1 and TIMP2 through LDR irradiation could influence activation of TGF-β3 via a change in the availability of specific MMPs/ADAMs/ADAMTSs. 

Inhibition of MMP/ADAMs in ICCM before transfer to reporter cells led to a surviving fraction after 0.1 Gy that was consistent with HRS, and lower than that of cells that had been treated with ICCM without the inhibitor. This points to an involvement of at least one member of either the MMP or ADAM protein families in the TGF-β3 dependent mechanism of HRS removal. Of the MMPs that were detected either as proteins in sEVs or as mRNA in whole cells, MMP2 [16], MMP13 [18] and MMP14 [41] are known to activate TGF-β1 or TGF-β2, and could possibly activate TGF-β3 considering their similarity. In addition, MMP9 has been shown to activate TGF-β3 [17] and is another possible candidate, although it was not detected in either of our analyses. While TIMP1 and TIMP2 have similar effects on most MMPs, they differ in their action on MMP14, which is inactivated by TIMP2, but not by TIMP1 [42]. Therefore, we conclude that at least one protein in the MMP or ADAM families is responsible for activation of TGF-β3 and removal of HRS, and consider MMP2, MMP9, MMP13 and MMP14 to be likely candidates.

### 3.3. Identification of Receptor

TGF-β3 normally activates the Smad 2/3 pathway through ALK5 after first binding to receptor TGF-βRII, which subsequently recruits and activates ALK5. Therefore, we first tested inhibition of ALK5 using ALK5 inhibitor SB431542, which did not alter the effect of TGF-β3 on HRS. SB431542 is also an inhibitor of ALK4 and ALK7 [43], so the involvement of either of these receptors with TGF-β3 in modifying HRS appears to be unlikely. We then tested inhibitor K02288, a blocker of ALK1, ALK2, and to some degree ALK6. In this case, a full HRS response was seen. To exclude involvement of ALK2 and ALK6, LDN193189, a blocker of ALK2, ALK3, and ALK6 was also tested, and we conclude that ALK1 is the most likely receptor mediating the effect of TGF-β3 on HRS. TGF-β3 has previously been shown to be a ligand to ALK1 [30]. However, to our knowledge, this is the first study to demonstrate a cellular function of TGF-β3 to depend on ALK1 activity independently of ALK5 and TGF-βRII.

ALK5 and ALK1 activate different Smad pathways. While activated, ALK5 phosphorylates Smad 2/3, whereas activated ALK1 phosphorylates Smad 1/5/8 [44,45]. TGF-β1 has been shown to regulate the activation state of the endothelium via a fine balance between ALK5 and ALK1 signaling. Whereas the TGF-β1/ALK5 pathway led to inhibition of cell migration and proliferation [46], the TGF- β1/ALK1 pathway induced endothelial cell migration and proliferation. However, another group found that ALK1 signaling inhibited the proliferation and migration of endothelial cells [47]. Increased ALK1/ALK5 ratio has also been seen to correlate with MMP13 expression in age-dependent osteoarthritis. During angiogenesis, ALK1 and ALK5 signaling pathways in endothelial cells have been found to play a crucial role in determining vascular endothelial properties [48].

The type II TGF-β receptors function primarily as binding receptors. On binding their ligand, type II receptors associate with and phosphorylate type I receptors, which in turn activate downstream Smad proteins. TGF-βRII is the only type II receptor that has been shown to interact with TGF-β1 or TGF-β3 [49]. TGF-β1 and TGF-β3 use ALK5 as a type I receptor in most cells after first binding to TGF-βRII, but ALK1 has also been shown to be activated by TGF-β1 in complex with TGF-βRII in endothelial cells [48]. The effect of TGF-β3 on HRS did not appear to be prevented by inhibiting TGF-βRII, suggesting that TGF-β3 either binds ALK 1 directly or uses another type II receptor. However, TGF-βRII has no close human relatives [15]. Park et al. found that TGF-βRII was not relevant to ALK1 signaling, as TGF-βRII-conditional deletion did not affect vessel morphogenesis in hereditary hemorrhagic telangiectasia type 2 (HHT2) in mice. They suggested that impaired signaling through TGF-β superfamily ligands outside of the TGF-β subfamily was involved in HHT2 pathogenesis [49]. 

### 3.4. Competition between Receptors and Possible Alternative to TGF-β3 Activation

In the present study, the effect of TGF-β3 on HRS appears to only involve ALK1, as inhibition of ALK5 did not affect the removal of HRS by TGF-β3. Interestingly, adding ALK5 inhibitor without TGF-β3 also led to removal of the HRS response. However, when anti-TGF-β3 neutralizing antibody was added concurrently, the HRS response was retained. We interpret these findings as ALK5 and ALK1 competing for TGF-β3, suggesting a low presence of active TGF-β3 even in unirradiated T-47D cells, which is normally scavenged by ALK5. When the function of ALK5 is inhibited, the amount of TGF-β3 available for ALK1 binding reaches a level where ALK1 is activated and HRS is removed.

In light of these results, ALK5 inhibition, as an alternative to activation of TGF-β3, emerges as a possible mechanism of removal of HRS in reporter cells. In the proteomic analysis, FKBP4 was found to be upregulated in sEVs from LDR primed cells compared to controls. FKBP4 could potentially function as an indirect inhibitor of ALK5 through its association with HSP90, a known stabilizer of ALK5 [31,32,33,34]. However, the radiosensitivity of T-47D cells was not altered after pretreatment with FKBP4, an indication that it is not involved in the removal of HRS through LDR priming. 

### 3.5. Clinical Perspectives

The results presented in this study highlight the role of TGF-β3 in the mechanism of HRS removal in T-47D cancer cells, thereby decreasing their radiosensitivity. In cancer radiotherapy, it is crucial to eradicate malignant cells while simultaneously maintaining healthy tissue. Here, the low dose region of IR is of concern, especially due to repeated low-dose irradiations of healthy tissue in the radiation field, through fractionation. HRS is observed in healthy as well as cancerous cell lines [1,2,3,4,5,6]. It is, therefore, of interest to investigate modification of low-dose radiosensitivity not only in cancer cell lines, but also in normal cells. An accurate and extensive understanding of these mechanisms in both types of tissue could potentially permit an improvement of the therapeutic ratio by manipulating their response to low-dose IR. 

## 4. Materials and Methods

### 4.1. Cell Culture

Cells of the human breast cancer cell line T-47D [50] (purchased from ATCC LGC Standards AB, Se-501 17 Boras, Sweden) were grown as monolayer cultures in RPMI 1640 medium (Gibco, Rockwell, MD, USA) 5 mL in a 25 cm^2^ flask (Nunc A/S, Roskilde, Denmark), supplemented with 10% fetal calf serum (Euroclone, Milano, Italy), 2 mM l-glutamine, 200 units/liter insulin and 1% penicillin/streptomycin (all from Gibco, Rockwell, MD, USA) as previously described [51]. Cells were recultivated twice per week with an additional medium change once per week. The T-47D cell line was chosen on the basis of its pronounced HRS response [7], as well as extensive previous research regarding removal of HRS in this cell line from our laboratory [7,21,22,24,35,36,52,53,54,55]. 

### 4.2. Irradiation

In the MMP/ADAM inhibitor (Figure 5), the cells were irradiated with a 220 kV X-ray source with hard filtering (1.52 mm Al and 2.60 mm Cu filters), at a dose rate of 22.5 Gy/h. In the sEV transfer (Figure 1), receptor (Figure 6 and Figure 7) and FKBP4 (Appendix A) experiments, the cells were γ-irradiated as described previously with a high dose rate of 20–25 Gy/h [22]. All LDR priming was conducted as previously described, but because of ^60^Co-decay, the low dose rate used in the present experiments was 0.15–0.2 Gy/h, compared to 0.3 Gy/h in previous studies. The total irradiation time was 1 h, so the total dose in all LDR irradiations was 0.15–0.2 Gy. This dose rate had the same effect on the HRS response as 0.3 Gy/h [24]. We have previously shown that T98G and T-47D cells exposed to 0.06–0.3 Gy/h (LDR) for 1 h permanently lose HRS. sEVs and ICCM were isolated from cells that had been LDR irradiated 1–3 months previously. The cells were exponentially growing (not confluent) at all times during the experiments. The temperature was kept at 37 °C during all procedures.

### 4.3. sEV Isolation

Before isolation of sEVs, T-47D cells were washed with serum-free medium and grown for 24 h without serum before the medium was harvested and filtered through a 0.8 µm Minisart filter (Sartorius AG, Goettingen, Germany). sEVs were extracted using membrane affinity spin columns provided by exoEasy maxi kit (QIAGEN, Manchester, UK), according to the protocol provided by the manufacturer.

The isolation protocol was verified by scanning the sEV eluate using a Philips CM 120 Bio Twin at the Electron Microscopic Laboratorium, Institute of Oral Biology, University of Oslo.

### 4.4. Clonogenic Assay Pretreatments

T-47D cells were subjected to various pretreatments before assessment of radiosensitivity via the clonogenic assay. Recombinant TGF-β3, TGF-β3 neutralizing antibody and TGF-βRII neutralizing antibody were purchased from R&D (243-B3, AF-243-NA and AF-241-NA; R&D Systems, Minneapolis, MN, USA), ALK1 inhibitor K02288 from Selleckchem (SMS-group, Rungsted, Denmark) and ALK5 inhibitor SB 431542 and iNOS inhibitor 1400 W from Sigma-Aldrich (Saint Louis, MO, USA). MMP/ADAM inhibitor TAPI-2 was purchased from Abcam (Cambridge, Great Britain). 

For the sEV transfer experiments in Figure 1, the sEV eluate was added to cell culture medium, which was transferred to the unirradiated reporter cell flasks and left on the cells for 24 h, before the cells were seeded in fresh medium for colony formation. The cells were then challenge γ-irradiated after another 18–19 h. As illustrated in Figure 1c, 1 µg/mL neutralizing anti-TGF-β3 antibody or 10 µM iNOS inhibitor 1400 W was added together with the sEV eluate for 24 h.

As shown in Figure 5, LDR primed T-47D cells were cultured in RPMI 1640 medium with 10 µM TAPI-2 for 24h. The medium was then filtered through a 0.2 µm Minisart filter (Sartorius AG, Goettingen, Germany) to remove cells and transferred to unirradiated T-47D reporter cells. Twenty-four hours after medium transfer, cells were seeded for the clonogenic survival assay in fresh medium without pretreatments. X-irradiation was performed 18–20 h after seeding.

The receptor experiments were conducted in two different ways. For the full survival curves in Figure 6, the cells were exposed to the pretreatment for 24 h, before being seeded for colony formation in fresh medium 18–19 h before irradiation. In these experiments, the controls were also exposed to the pretreatment. For the data presented in Figure 7, the pretreatments were added as the cells were seeded for colony formation 16–20 h before irradiation and maintained until the cells were fixated after about 3 weeks. In these experiments, the controls were not exposed to the pretreatment. The plating efficiency of unirradiated cells seeded in medium with 10 pg/mL recombinant TGF-β3 or 2 µg/mL anti-TGF-β3 (maintained until fixation) was 0.96 ± 0.03 and 0.93 ± 0.01 compared to the plating efficiency of untreated cells, respectively.

### 4.5. Cell Survival

The cells were trypsinized and counted in a Bürker chamber before seeding. For the TAPI-2 experiments, 4 parallel flasks were seeded for each dose, and 8 for unirradiated controls. For the sEV, receptor and FKBP4 survival experiments, 5 parallel flasks were seeded for each dose, and 10 for controls. The cells for the controls and irradiated flasks were seeded from the same dilution. After 14–21 days of incubation without medium change or re-cultivation, the cells were fixated and stained with methyl blue (Waldeck GmbH & Co, Münster, Germany). The fixated cells were counted manually using an illuminated magnifier (Gerber Instruments, Effretikon, Switzerland) and a light microscope (Nikon Instruments, Tokyo, Japan) at 10X magnification. Colonies with more than 50 cells were scored as survivors, and the surviving fraction was calculated as the number of colonies after irradiation relative to the number of colonies in the unirradiated control. In order to account for increasing multiplicity per colony-forming unit during the time interval from seeding and irradiation, an extra flask was seeded for each experiment. This flask was fixated at the time of irradiation, and the multiplicity was counted. The mean value was calculated and used for corrections according to a formula previously published [56].

### 4.6. Nano-LC LTQ Orbitrap Mass Spectrometry

Proteomic analysis was performed in two separate experiments, where sEVs were isolated from LDR primed and unirradiated control T-47D cells. There were three samples per group. Each Coomassie G-250 stained SDS-PAGE gel lane was cut into 12 slices, and each of them in-gel digested using 0.1 µg of trypsin in 25 µL of 50 mM ammonium bicarbonate, pH 7.8. After micro purification using µ-C18 ZipTips (Millipore, Oslo, Norway), the peptides were dried in a SpeedVac and dissolved in 10 µL 1% formic acid, 5% acetonitrile in water. Half of the volume was injected into an Ultimate 3000 nanoLC system (Dionex, Sunnyvale, CA, USA) connected to a linear quadrupole ion trap-orbitrap (LTQ-Orbitrap XL) mass spectrometer (ThermoScientific, Bremen, Germany) equipped with a nanoelectrospray ion source. For liquid chromatography separation, an Acclaim PepMap 100 column (C18, 3 µm beads, 100 Å, 75 μm inner diameter) (Dionex, Sunnyvale CA, USA) capillary of 50 cm bed length was used. The flow rate was 0.3 μL/min, with a solvent gradient of 7 % B to 35 % B in 110 minutes. Solvent A was aqueous 0.1 % formic acid, whereas solvent B was aqueous 90 % acetonitrile in 0.1 % formic acid. The mass spectrometer was operated in the data-dependent mode to automatically switch between Orbitrap-MS and LTQ-MS/MS acquisition. Survey full scan MS spectra (from m/z 300 to 2000) were acquired in the Orbitrap with the resolution R = 60,000 at m/z 400 (after accumulation to a target of 1,000,000 charges in the LTQ). The method used allowed for the sequential isolation of up to the seven most intense ions, depending on signal intensity, for fragmentation on the linear ion trap using collision-induced dissociation at a target value of 10,000 charges. Target ions already selected for MS/MS were dynamically excluded for 60 s. The lock mass option was enabled in MS mode for internal recalibration during the analysis. Other instrument parameters were set as previously described [57].

### 4.7. Protein Identification and Quantification

Protein identification and label-free MS1 quantification were performed using PEAKS® Xpro [58] (Bioinformatics Solutions Inc., Waterloo, ON, Canada). Analysis parameters for the quantification of differentially detected proteins are provided in Appendix A. 

The interactions between differentially detected proteins and potentially relevant query proteins were analyzed using STRING [28].

### 4.8. mRNA Analysis

Results from a previously published transcriptomic analysis of LDR primed and control T-47D cells were re-analyzed and compared with the results from the proteomic analysis in the current study [24]. Previously, unirradiated and LDR primed T-47D cells were harvested 2 months after irradiation. RNA purification was performed using a Qiagen RNAeasy minikit (Qiagen 74104, Qiagen, Germany), and two-color microarray-based gene expression analysis was performed using a 44 K human Whole Genome Oligo microarray kit from Agilent Technologies. Images were scanned on the Agilent Technologies Scanner G2505B US22502537 and quantified using Agilent Feature Extraction Software (version 9.1.3.1). Bioinformatic analysis was performed using the Bioconductor package LIMMA [59]. Differentially transcribed genes between LDR primed and unirradiated T-47D cells were identified using a linear model with a modified t-test comparing the two groups. Raw and normalized data from this study are available from Gene Expression Omnibus (GEO) under the accession number GSE41483.

### 4.9. Statistical Analysis

The proteomic data in Figure 2 and Figure 3 were analyzed using one-way ANOVA comparing sEVs from LDR primed and unirradiated control cells, with a significance level of 0.05. For the mRNA data in Figure 2 and Figure 3, a modified *t*-test comparing LDR primed and unirradiated controls was used, with T-47D cells HDR irradiated 24 hours prior as a reference [27]. Log-odds of whether the genes were differentially transcribed (B-statistics) were used to rank the genes. *p*-values were corrected for multiple testing using Benjamini and Hochberg false discovery [60].

The surviving fractions in Figure 5, Figure 7 and Appendix A were analyzed using one-way ANOVA with a significance level of 0.05, followed by a post hoc Tukey’s HSD test.

## 5. Conclusions

Our current paper investigated the TGF-β3 dependent mechanism of removal of HRS through LDR priming of cells. We found that TGF-β3 was secreted in similar amounts in sEVs from LDR primed and control cells and did not detect TGF-β1 or TGF-β2 in sEVs. Whereas sEVs from unirradiated control cells did not affect the radiosensitivity of reporter cells, sEVs from LDR primed cells removed HRS upon subsequent challenged irradiation. We conclude that TGF-β3 is secreted in an inactive complex with LAP from cells regardless of priming, but with an activating factor from cells that have been LDR primed. Our results show that HRS is not removed from reporter cells if proteins from the MMP and ADAM protein families are inhibited, identifying these as probable activators of TGF-β3 in this mechanism. 

We demonstrated for the first time a cellular function of TGF-β3 to depend on the receptor ALK1 in the removal of HRS. This function was independent on ALK5 and TGF-βRII. We also revealed a competition between ALK1 and ALK5 for binding TGF-β3, where ALK5 had higher affinity for the ligand, but ALK1 mediated the radioprotective effect.

## Figures and Tables

**Figure 1 ijms-23-08147-f001:**
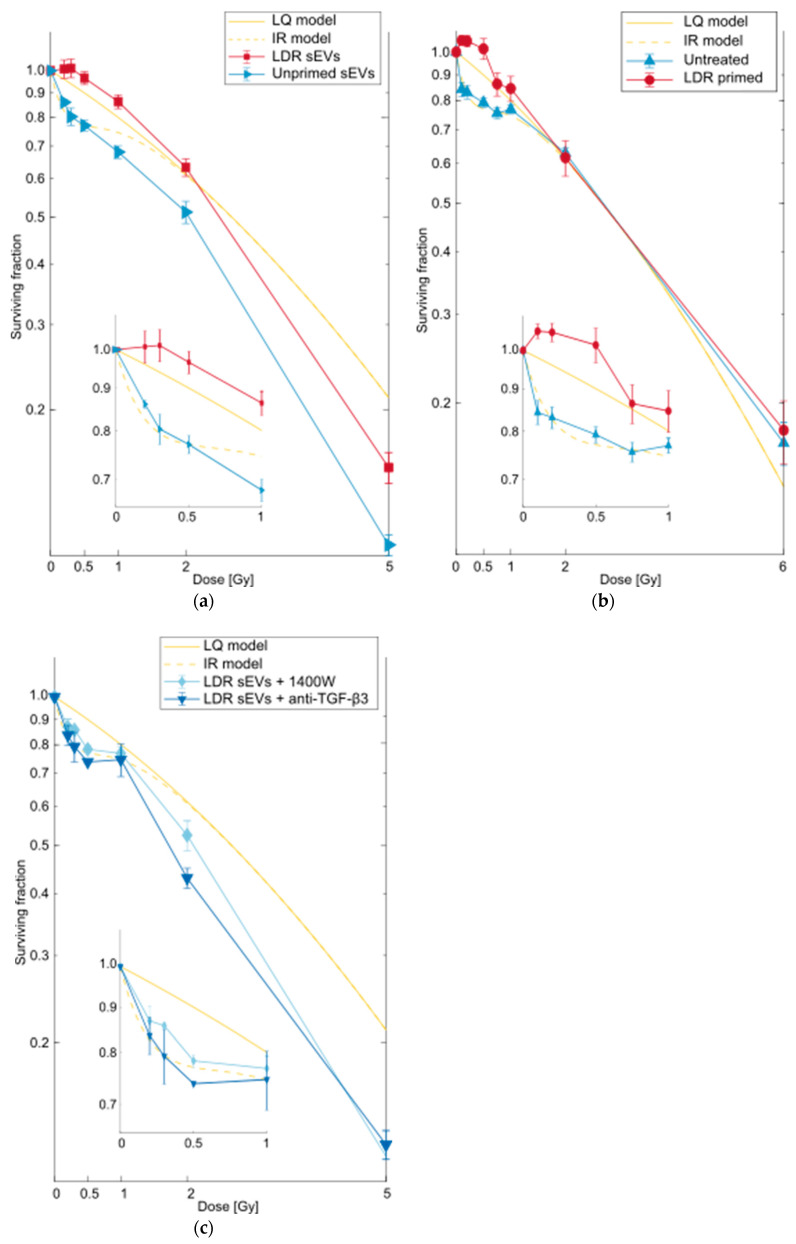
Survival curves for T-47D cells after pretreatment with small extracellular vesicles (sEVs). (**a**) Pretreatment with sEVs from low dose rate (LDR) primed T-47D cells (■) removed the hyper-radiosensitive (HRS) response to subsequent challenge irradiation in T-47D reporter cells. Pretreatment with sEVs from unirradiated controls (►) had no effect on the radiosensitivity of reporter cells. (**b**) Replot from [7]. Previously untreated T-47D cells (▲) exhibit the HRS response to low doses of ionizing radiation (IR). T-47D cells that were primed with a dose of 0.3 Gy delivered at an LDR of 0.3 Gy/h (●) lost HRS. (**c**) Pretreatment with inducible nitric oxide synthase (iNOS) inhibitor 1400 W (◆) or neutralizing antibody to transforming growth factor β3 (TGF-β3) (▼) together with sEVs from LDR primed cells, negated the effect of the sEVs of abolishing the HRS response in reporter cells. LQ model and IR model: linear-quadratic model [25] fit and induced repair model [26] fit, respectively, for untreated T-47D cells. Surviving fractions are given as error-weighted means of three separate experiments, each with five biological replicates. Error bars represent standard error of the mean (SEM). Note that the surviving fractions were calculated relative to the plating efficiency of control cells, which were also exposed to the pretreatments.

**Figure 2 ijms-23-08147-f002:**
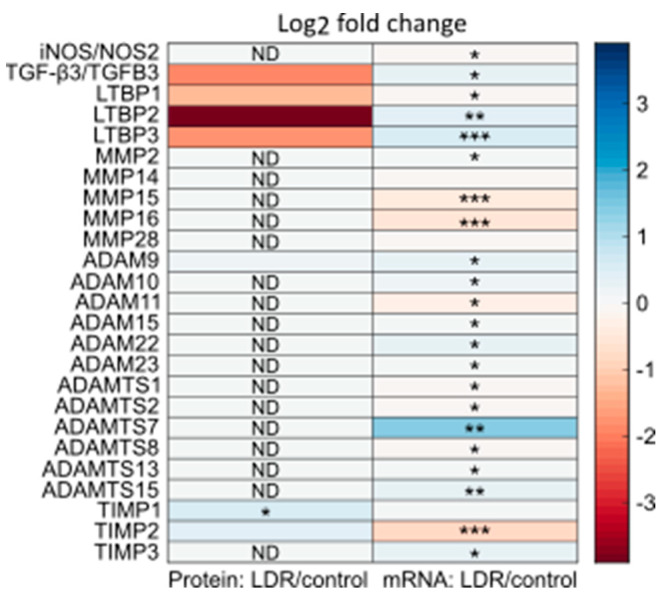
Left column: log2-fold change in detection of proteins of potential relevance in the removal of HRS by LDR irradiation, between sEVs from LDR primed T-47D cells compared to sEVs from unirradiated controls. Right column: log2-fold change in mRNA detection of corresponding genes between LDR primed and unirradiated T-47D whole cells. Proteins were selected based on results from previous experiments, because of relation to the TGF-β protein family, or as potential activators from the matrix metalloproteinase (MMP) and disintegrin and metalloproteinase (ADAM) families and their inhibitors, tissue metalloproteinase inhibitors (TIMP). ND = not detected. Proteomic data: highest significance out of two separate experiments, each with three biological replicates. Statistical analysis: one-way ANOVA for LDR irradiated against unirradiated control. mRNA data: average of four biological replicates. Statistical analysis: modified *t*-test [27]. * *p* < 0.05, ** *p* < 0.01, *** *p* < 0.001.

**Figure 3 ijms-23-08147-f003:**
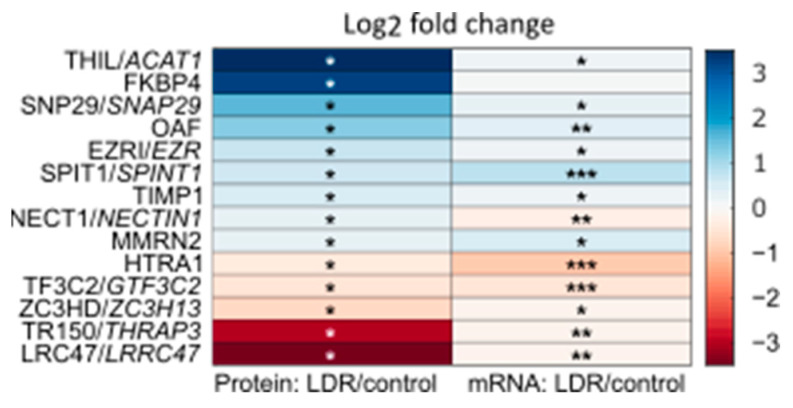
Left column: log2-fold change of protein detection in sEVs from LDR primed T-47D cells compared to sEVs from unirradiated controls. Right column: corresponding mRNA detection in LDR primed whole T-47D cells compared to unirradiated controls. Only proteins that were significantly up- or downregulated in LDR sEVs in either of two experiments are included in the plot. Proteomic data: highest significance out of two separate experiments, each with three biological replicates. Statistical analysis: one-way ANOVA for LDR irradiated against unirradiated control. mRNA data: average of four biological replicates. Statistical analysis: modified *t*-test [27]. * *p* < 0.05, ** *p* < 0.01, *** *p* < 0.001.

**Figure 4 ijms-23-08147-f004:**
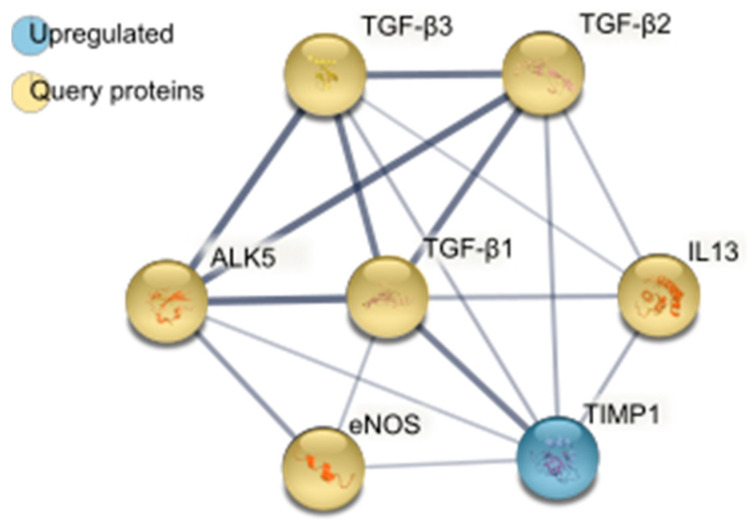
STRING [28]-generated protein–protein interaction (PPI) analysis map of metalloproteinase inhibitor 1 (TIMP1) with query proteins of potential relevance for removal of HRS by LDR irradiation. Query proteins were selected based on results from previous experiments, or because of a known relation to the TGF-β protein family. Only query proteins with a predicted interaction with TIMP1 are shown. Thickness of the lines indicates interaction confidence, as determined by STRING according to the Benjamini–Hochberg procedure [28].

**Figure 5 ijms-23-08147-f005:**
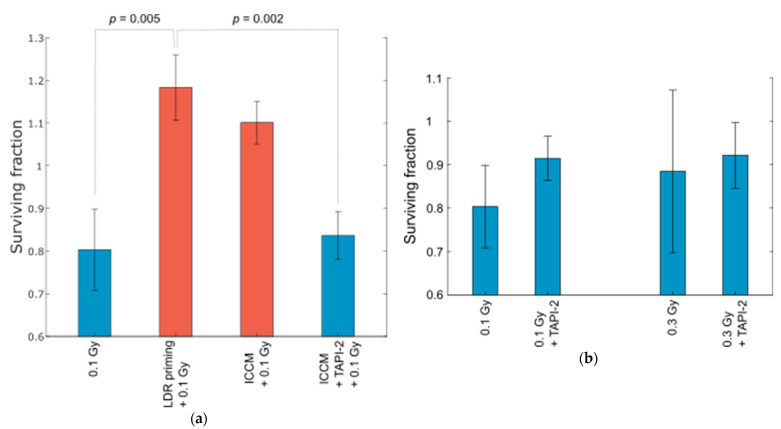
Survival of T-47D cells after pretreatment with MMP/ADAM inhibitor. (**a**) As previously demonstrated, LDR priming of T-47D cells removed HRS, shown here by an increase in the surviving fraction after 0.1 Gy x-irradiation. Pretreatment with irradiated cell conditioned medium (ICCM) had a similar effect on the reporter cells. However, pretreatment with MMP/ADAM inhibitor TAPI-2 negated the effect of ICCM alone, resulting in a surviving fraction consistent with HRS. Note that the surviving fractions were calculated relative to the plating efficiency of control cells, which were also exposed to the Pretreatments. (**b**) Pretreatment with TAPI-2 alone did not significantly influence the surviving fraction of T-47D cells after 0.1 or 0.3 Gy. Surviving fractions are given as error-weighted means of three separate experiments, each with four biological replicates, except for ICCM + TAPI-2 + 0.1 Gy for which five separate experiments with four biological replicates were performed. Error bars represent SEM. Statistical analysis: One-way ANOVA with post hoc Tukey’s honest significant difference (HSD).

**Figure 6 ijms-23-08147-f006:**
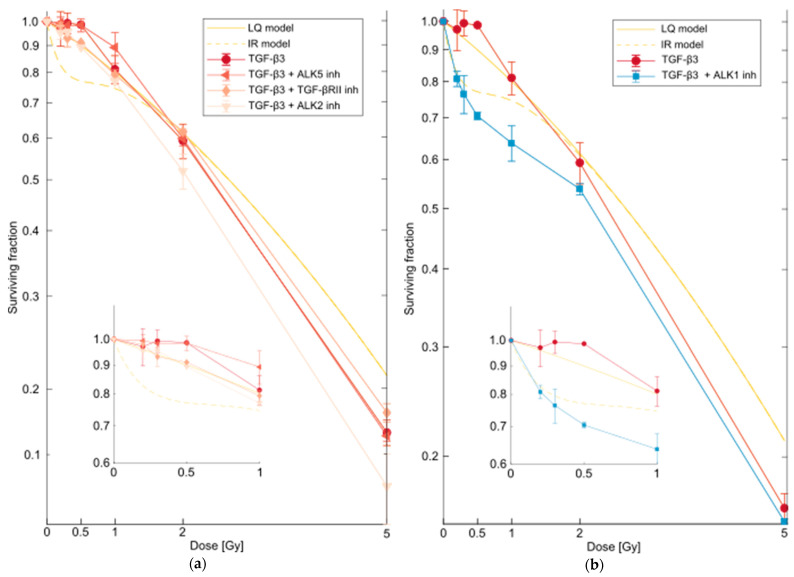
Survival curves for T-47D cells after pretreatment with TGF-β3 and inhibitors of relevant receptors. (**a**) As previously demonstrated, pretreatment with TGF-β3 (●) removed the HRS response in T-47D cells. Pretreatments with inhibitors of activin receptor like kinase 5 (ALK5) (◀), TGF-βRII (◆), or ALK2 (▼) together with TGF-β3 did not influence the ability of TGF-β3 to remove the HRS response. (**b**) Pretreatment with inhibitor of activin receptor like kinase 1 (ALK1) together with TGF-β3 (■) negated the effect of TGF-β3 alone (●) on the removal of HRS in T-47D cells. LQ model and IR model: linear-quadratic model [25] fit and induced repair model [26] fit, respectively, for untreated T-47D cells. Surviving fractions are given as error-weighted means of three separate experiments, each with five biological replicates. Error bars represent SEM. Note that the surviving fractions were calculated relative to the plating efficiency of control cells, which were also exposed to the pretreatments.

**Figure 7 ijms-23-08147-f007:**
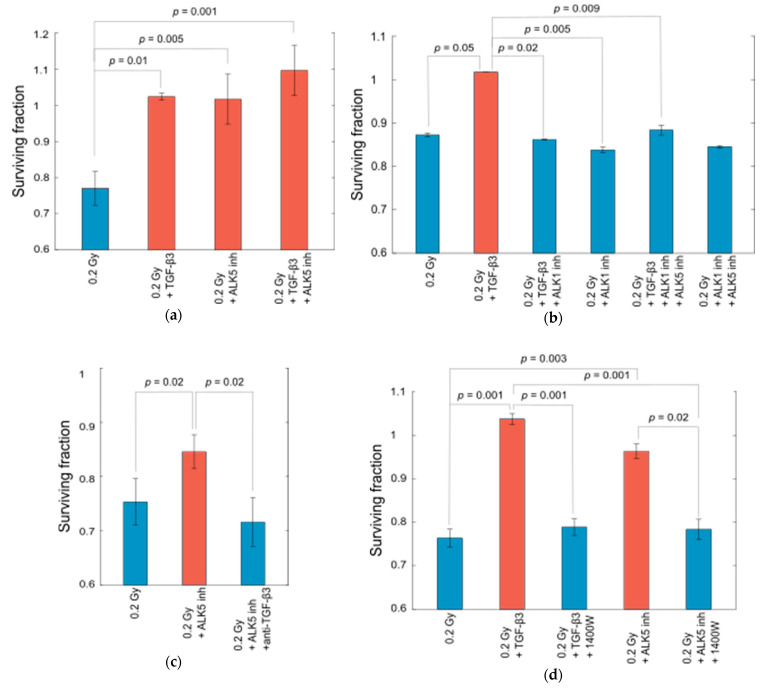
Surviving fraction of T-47D cells after 0.2 Gy and pretreatment with inhibitors of different receptors. (**a**) As previously demonstrated, pretreatment with TGF-β3 removed HRS in T-47D cells, shown here by an increase in the surviving fraction after 0.2 Gy γ-irradiation. Pretreatment with inhibitor of ALK5 resulted in a similar effect, both alone and in combination with TGF-β3. (**b**) Pretreatment with inhibitor of ALK1 negated the ability of TGF-β3 to remove HRS in T-47D cells when added together, and did not influence the surviving fraction when added alone. Pretreatment with inhibitor of ALK5 in combination with inhibitor of ALK1 did not influence the latter’s effect on the surviving fraction with or without addition of TGF-β3. (**c**) Removal of HRS by inhibition of ALK5 was negated by combined pretreatment with neutralizing antibody to TGF-β3. (**d**) Pretreatment with iNOS inhibitor 1400 W in combination with either TGF-β3 or inhibitor of ALK5 restored HRS to T-47D cells. Surviving fractions are given as error-weighted means of two (b) or three (a, c, d) separate experiments, each with five biological replicates. Error bars represent SEM. Statistical analysis: One-way ANOVA with post hoc Tukey’s HSD. Note that the surviving fractions were corrected against the plating efficiency of control cells that had not been subjected to pretreatment.

## Data Availability

This study used microarray data that were previously deposited to Gene Expression Omnibus under the accession number GSE41483. Proteomic data from this study have been deposited to the PRIDE database under the accession number PXD035010. Data from all clonogenic assays in this study have been deposited to Zenodo and assigned the identifier https://doi.org/10.5281/zenodo.6685318.

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
