# Peer review of "Low-Dose-Rate Radiation-Induced Secretion of TGF-β3 Together with an Activator in Small Extracellular Vesicles Modifies Low-Dose Hyper-Radiosensitivity through ALK1 Binding"

_ijms, 2022, doi:10.3390/ijms23158147_

Round 1

Reviewer 1 Report

The article entitled " Low dose rate radiation induced secretion of TGF-β3 together 2 with an activator in small extracellular vesicles modifies low 3 dose hyper-radiosensitivity through ALK1 binding « highlights the role of  the transforming growth factor β3 in the abolition of the HRS effect.

Article is of interest but some point should be improved.

My first remark concerns the cell line used in this study. Why did the authors realized their experiments on T4-7D (a breast cancer cell line). This point has to be explained 

Authors indicated that it is important to understand the difference HRS mechanism between normal and tumoral cell. It is important to understand this effect on healthy cells to avoid it and also on tumoral cells to improve it. It could be of interest to add some sentence about this concern in introduction and discussion and also validate some of these observations on normal cells.

Following sentences are not important : « TGF-β1 null mice 50 exhibit multifocal inflammatory disease, and die within three weeks of birth TGF-β2 null mice exhibit a wide range of developmental effects and perinatal mortality and TGF-β3 null mice exhibit a cleft palate resulting in an inability to suckle and death within 24 hours after birth. In addition to their role in embryogenesis « 

Authors have focused their attention on the processus of TGFb3 activation.

« We show that the factors responsible for removing HRS are released from LDR irradiated cells in small extracellular vesicles (sEVs), and that sEVs contain TGF-β3 regardless of irradiation. « 

I’m wondering on this sentence because authors indicate above that HRS is abolished with LDR irradiated conditioned media from untreated cells.

Does sEVs from unirradiated cells contains also TGFb3?

Figure 3 should be displaced in suppl data.

L157: differenetially should be corrected.

2.3

It can be of interest to discriminate involvement of Adam and MMP by using GM6001 

Fig 6a authors have to realize experiments with LDR and compare to sEV because they compare LDR, 0,1 Gy to ICCM + TAPI -2 (sEV composition is different from ICCM)

Experiments with recombinant MMP14 have to be detailed. Did authors used a truncated active form without TM domain?

In this context authors should transfect their cells to express full size MT1 MMP because results could be different.

Focus on this MMP should be explained because we don’t know why they studied it.

It could be of interest to use an MMP/ADAM protein array to precise this point

Authors could also check involvement of αvβ6 and αvβ8

« HSP90 has in turn been shown to interact with and 259 stabilize ALK5[35–37], and we therefore considered the possibility that an increase in the 260 FKBP4 concentration in LDR primed cells led to a decrease in the amount of HSP90 261 «  

Experiments concerning HSP90 expression should be done to validate this hypothesis by immunoprecipitation assays by exemple.

Discussion

1.2

Smad signaling should be also studied by western blotting to precise and validate signaling pathways

Clinical perspectives should be proposed.

Reviewer 2 Report

The author revealed the cellular function of TGF-β3 to depend on the receptor ALK1 in the removal of HRS. This function was independent of ALK5 and TGF-542 βRII. They also reveal a competition between ALK1 and ALK5 for binding TGF-β3, where 543 ALK5 has higher affinity for the ligand, but ALK1 mediates the radioprotective effect.  Albeit, this study provides some guidance for the radiosensitivity-related field, I still have some suggestions.

1, All figures are highly professional, and the authors should guide the readers to the meaning of the images appropriately; otherwise, it is likely to cause misunderstandings. Therefore, I suggest that the author consider revising these figure legends again.

2, In figure 2, the author used Log2-fold change in the detection of proteins of potential relevance in the removal of HRS by LDR irradiation. Since they used breast cancer cell line T-47D for the current study, therefore, it would be very interesting, if the author can predict the survival outcome of MMPs/ADAMs/ADAMTS in breast cancer patients via TCGA or Kaplan-Meier plotter database(PMID: 34309564, 32064155, 34202528).

3, The authors gave a general answer on gene expression by performing 44 K human Whole Genome Oligo microarray kit from Agilent Technologies (GSE41483). To further confirm their data, please perform pertinent bioinformatic analyses and investigate their data on DNA methylation (https://biit.cs.ut.ee/methsurv/) (PMID: 29264942, 34834441, 35741647).

4, The author should use other statistical analyses such as ANOVA to calculate the P-value for three or more groups of data. For example, in Fig 6 and 8, please add the correct P-value either in the result or figure legends, and also update the “Statistical Analysis” of Method during further revision (PMID: 32635204, 34329194, 33805706). 

5,  There are few typo issues for the authors to pay attention. Please unify the writing of scientific terms. “Italic, capital” ? make it consistent throughout the whole manuscript. For example, Page 11, Line 347, “1.2” Identification of receptor should be corrected as “3.3” Identification of receptor.

Round 2

Reviewer 1 Report

Authors have answered to my requested questions.

Reviewer 2 Report

Authors addressed the most of reviewers' concerns; now they are more readable and make sense.